# Evaluation of the Spatial Effect of Network Resilience in the Yangtze River Delta: An Integrated Framework for Regional Collaboration and Governance under Disruption

Lei Che [1], Jiangang Xu [1], Hong Chen [2,*], Dongqi Sun [3], Bao Wang [4], Yunuo Zheng [5], Xuedi Yang [6] and Zhongren Peng [7]

1. School of Architecture and Urban Planning, Nanjing University, Nanjing 210093, China
2. Faculty of Geomatics, Lanzhou Jiaotong University, Lanzhou 730070, China
3. Institute of Geographic Sciences and Natural Resources Research, University of Chinese Academy of Sciences, Beijing 100101, China
4. Lanzhou Information Center, Northwest Institute of Eco-Environment and Resources, Chines Academy of Sciences, Lanzhou 730000, China
5. Urban-Planning & Design Institute, Zhejiang University, Hangzhou 310030, China
6. College of Earth and Environmental Sciences, Lanzhou University, Lanzhou 730000, China
7. International Center for Adaptation Planning and Design, College of Design, Construction and Planning, University of Florida, Gainesville, FL 32611, USA
* Correspondence: chenhong@edu.lzjtu.edu.cn

**Abstract:** Public health emergencies are characterized by significant uncertainty and robust transmission, both of which will be exacerbated by population mobility, threatening urban security. Enhancing regional resilience in view of these risks is critical to the preservation of human lives and the stability of socio-economic development. Network resilience (NR) is widely accepted as a strategy for reducing the risk of vulnerability and maintaining regional sustainability. However, past assessments of it have not sufficiently focused on its spatial effect and have overlooked both its internal evolution characteristics and external threats which may affect its function and effectiveness. Therefore, we used the Yangtze River Delta Region (YRDR) as a case study and conceptualized an integrated framework to evaluate the spatial pattern and mechanisms of NR under the superposition of the COVID-19 pandemiv and major holidays. The results indicated that the topology of a population mobility network has a significant effect on its resilience. Accordingly, the network topology indexes differed from period to period, which resulted in a decrease of 17.7% in NR. For network structure, the Shanghai-Nanjing and Shanghai-Hangzhou development axes were dependent, and the network was redundant. In the scenario where 20% of the cities were disrupted, the NR was the largest. Furthermore, the failure of dominant nodes and the emergence of vulnerable nodes were key factors that undermined the network's resilience. For network processes, NR has spatial effects when it is evolute and there is mutual inhibition between neighboring cities. The main factors driving changes in resilience were found to be GDP, urbanization rate, labor, and transportation infrastructure. Therefore, we propose a trans-scale collaborative spatial governance system covering "region-metropolitan-city" which can evaluate the uncertain disturbances caused by the network cascade effect and provide insights into the sustainable development of cities and regions.

**Keywords:** network resilience; interrupt simulation; spatial effect; regional governance; Yangtze River Delta region

## 1. Introduction

Network resilience refers to the ability to withstand shocks and to restore, maintain, or improve the characteristics and critical functions of a system, as well as providing an important supporting role to each unit within a region [1,2]. As COVID-19 continues to spread throughout the world, it is having a significant impact on economic and social

development [3,4]. Since it is extremely transmissible, restricting the mobility of the population has allowed the spread of the disease to be effectively contained while also reducing urban connectivity [5,6]. The "blocking cities" measures have reduced external risks [7]. Meanwhile, the incident has had an adverse effect on regional connections as well as routine operations [8]. The global spread of COVID-19 was an unconventional disaster. As such, it may be hypothesized that urbanization will increase vulnerability [9]. When not supported by a healthy and safe environment, such an event can even produce long-term "shock" and "aftereffects" within a short period of time [10]. With innovations in risk management, such as dynamic adaptation, comprehensive promotion, and multi-stakeholder collaboration, urban resilience is becoming a hot topic in disaster prevention, risk mitigation, and urban planning. To promote better regional sustainable development, it may be beneficial to make cities more resilient to shocks.

As urbanization is characterized by high density and mobility, it has broken through static administrative boundaries [11]. The mobility of the population has come to be characterized by scale, normality, dynamism, and, most importantly, a complexity of behaviors [12]. There has been a precipitous increase in transmission speed, resulting in an increased risk of security incidents. Therefore, it is important to explore population mobility using the theory of mobility space and from a network perspective in order to assess the regularity of spatial movement at a city or regional scale, as well as the dynamic characteristics of inter-city spatial connections. Urban network resilience can be used to evaluate regional resilience, which refers to the ability to maintain, improve, or restore the original performance and function following a shock [13]. Research in this field can be put into one of the following categories: concept and connotation, assessment of network structures, and simulations of shock resistance. Urban network systems can resist the impacts of acute external shocks while also adapting to internal pressures by improving the strength and agility of social, economic, ecological, engineering, and organizational relationships [14,15].

Much empirical research has been conducted on global and regional networks from the perspectives of enterprise location, information networks, and traffic connections. A great deal of literature has been produced about population mobility, especially the characteristics of intercity travel during specific periods, such as May Day, National Day, and the Spring Festival [16–19]. Network analysis has become the main method by which to study population mobility, and it has become a paradigm of how networks are represented. Moreover, the COVID-19 pandemic has led to more and more attention being paid to network resilience [20,21]. Frequent natural disasters and manufacturing emergencies can interrupt a network and affect the regular operation of the cities, with unpredictable consequences [22]. Generally, the investigation of targeted and random attacks on city nodes determines the degree of attenuation, the influencing factors, and the effectiveness of various optimization strategies for a specific network structure [23]. As a result, many cities are at risk of disasters or attacks, resulting in disruptions in urban communication. Thus, scenario simulations of networks under disruption can help predict whether urban networks will be able to resist potential risk to their operational capacity and capabilities when a public health emergency occurs.

However, there are also some research gaps in the previous literature, such as the absence of theoretical exploration. Although empirical studies have been important in understanding the characteristics and causes of population mobility, there has not been a unified paradigm of research due to the lack of systematic theoretical studies. Most studies have focused on economic development or disaster prevention and mitigation, while only a few have considered spatial relationships and effects. Urban network resilience should be given enough spatial consideration, which is different from the aforementioned regional resilience. additionally, we should focus on how to discuss the mechanisms of urban network resilience in sufficient depth, rather than building a network and analyzing its characteristics. Network science offers new perspectives on the complex networks in social, economic, and technological systems [24]. Network robustness and resilience are critical to the reduction of risk and mitigation of damage. Essentially, network resilience is the ability

of a system to maintain its essential functions when it faces internal disturbances or external changes [25]. Complex networks function as a result of the robustness of their structures, which can maintain connectivity if some nodes or edges are removed. Cities and regions are complex coupled systems which are characterized by complexity, diversity, nonlinearity, uncertainty, and multiscale nesting. Evaluations of network resilience offer an analytical perspective between humans and complex systems that are continually adapting [26].

Thus, we propose an integrated approach which utilizes both complex systems and spatial modeling analysis to identify the effects of a public health emergency disruption on regional network resilience. A multi-perspective network based on big data will help to understand spatial patterns and resilience under disturbance. The aims of our study are to: (1) conceptualize network resilience from the perspectives of a complex adaptive system (CAS) and a complex network (CN); (2) evaluate network resilience and its characteristics under different disruption scenarios; and (3) identify dominant and vulnerable nodes in urban linkage networks. Such efforts may answer the following scientific questions: What are the concepts, connotations, and characteristics of network resilience in the context of CAS and CN? What are the characteristics of population mobility network linkages over time? What are the main factors that affect network resilience under disruption due to external shocks? How can we reshape regional spatial structures for sustainable regional development while improving network resilience? We believe that the results of our study will provide new insights for policymakers when it comes to considering regional collaboration and identifying a trans-regional system of coordinated governance.

## 2. Conceptual Framework of Network Resilience from CAS

### 2.1. CAS Theory and Its Characteristics

CAS theory was developed by the Santa Fe Institute in the United States in the 1980s. It focuses on interactions between individuals and the environment, and thus represents a new means by which to view systems. There is an argument that the complexity of a system is caused by an individual's ability to adapt. The system in question will exhibit adaptive behavior in response to external disturbances. Diverse heterogeneous individuals also interact in autonomous and diverse ways which affect both their evolutionary paths and the structure of the system [27]. According to the theory, the transformation, evolution, and development of an organization are collective outcomes of the subject's active knowledge of the outside environment. There is a critical element of adaptive creation complexity, and one of its most important elements is adaptive subjectivity [28]. System evolution is fundamentally affected by the constant interactions between adaptive systems and their environments. Through the interaction process, the adaptive issue is raised to a new level, exhibiting a more complex structure and behavior [29].

### 2.2. Complex Network Resilience from CAS

Several articles have introduced assessment methods for resilience based on CAS, which authors have argued facilitate a better comprehension of the structure and operation mechanisms of an urban system [30,31]. The use of CAS theory to explore how CNs become resilient has not been reported in the literature. Thus, we define urban network resilience first as the dynamic nature of the structural characteristics of a network, as well as the internal adaptive adjustment process when the regional space system experiences external shocks (Figure 1). Complexity lies at the core of diversity, and the complexities associated with network resilience are derived both from the diversity of internal linkages and from the active adaptation of internal elements. Essentially, it consists of the interconnections among network nodes and the continuous adaptation of the system to external disturbances.

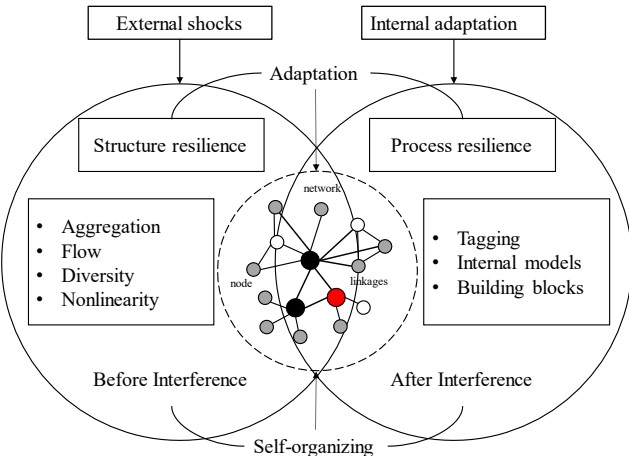

**Figure 1.** Conceptual framework for regional network resilience based on CAS.

When the resilience of a certain regional network is diminished, the function of the entire regional system will be affected. As a result, a previously stable structure and the ability to adapt to shocks and impacts will be altered, which will affect the entire regional network system. Due to structural changes, the existing structure will not have enough resistance and adaptability, and therefore, its resilience will be weakened in response to external shocks. In one sense, such a change would improve the overall network resilience (i.e., by reducing the redundancy of the offset network). On the other hand, such a change would weaken the resilience of the existing structure due to structural changes. Therefore, simulations of disruptions to urban networks can assist in predicting the capacity of regional urban networks to withstand potential risks, thus helping to reduce the adverse impact of disasters and improve regional resilience strategies.

According to our theoretical concepts, a complex network system is composed of two fundamental attributes: structure and process. These notions are different but interrelated. When a network system is subjected to external shocks, the first thing to occur is a change to its structural characteristics, i.e., the preparatory stages before the interruption. As a result, the system will experience self-regulation and adaptation, which is known as process resilience. This represents the adaptability and resilience of the system. Normally, when an external entity attacks a network node, the nodes and linkages suffer the first consequences. These network topology characteristics indicate the degree of impact of the node. After a period of self-recovery, the systems recover or approach their original state via self-regulation. Therefore, simulations of the topological indexes of network resilience, before and after an interruption, can reflect both the structural and process characteristics of the network to reveal the system resilience.

### 2.3. Characteristics and Mechanisms of Complex Network Resilience

CNs are used to analyze changes in network characteristics, while CASs are used to provide a qualitative understanding of the adaptation process of a system globally. Combined with the seven main attributes of CAS theory, the basic elements of the complex adaptive system are summarized below, and the characteristics of complex networks are deduced.

Aggregation, Flow, Diversity, and Nonlinearity are four basic features in CASs; they correspond to Centrality, Density, Diversity, and Aggregation in CNs, reflecting the structural characteristics of network resilience. The term "structural resilience" refers to the resilience problem caused by the topology of a network, with a particular focus on the physical and logical connections between network nodes, including the resilience of the nodes themselves, the resilience of the connections, and the overall resilience of the network. On the one hand, node and connection resilience emphasizes the destructive ability of these entities, while on the other, overall toughness emphasizes the ability of a network

to self-organize and coordinate itself, as well as to coordinate among its dimensions. The specific correspondence is as follows:

- Aggregation indicates that simpler subjects can emerge with more complex behavior through interactions among aggregates. The interactions between these subjects can give rise to higher-level subjects that generate new meta-agents through re-aggregation. This forms the hierarchy of a CAS. Using complex network centrality as a metric, it is possible to determine the hierarchy of a network of cities, which enables an urban network to accommodate a hierarchy of nodes. Generally, cities with high centrality in a network are highly distinguished, core components which enhance the cohesiveness and competitiveness of the network, but also increase the vulnerability of the network structure, since non-core cities are highly dependent on core cities.
- Flow refers to the continuous flow of resources between subjects or between the subject and the environment, including the flow of information, logistics, and capital. The nature of the flow will directly impact the evolution of the system. Network density is an important indicator that precisely reflects the complexity of the overall network. A higher density of nodes will result in a greater number of interconnections and a greater degree of mobility.
- Diversity results from the continuous adaptation of a subject; each new adaptation creates new possibilities for further interactions. Through a continuous cyclical process, significant differences between subjects occur, which contributes to the diversity of the system. This is one of the most important features of a CAS. Whenever the elements of a complex network are adversely affected by external factors, such as shocks, attacks, etc., in the process of communication through a particular path, the normal operation of the network can be maintained by quickly choosing other paths of communication.
- Nonlinearity is characterized by the idea that the whole is greater than the sum of the parts. Interactions between individuals are not simple causal relationships, but complex relationships based on mutual feedback within an adaptation process. Thus, CASs exhibit a wide range of properties and states, and nonlinearity is an inherent source of complexity. In complex networks, nonlinearity is represented by the average aggregation coefficient i.e., the degree of aggregation between neighboring nodes. Consequently, the process of aggregation also changes in nonlinear growth as a result of aggregation in the network, resulting in a complex relationship comprising mutual feedback among nodes.

Tagging, building blocks, and internal models are all reflections of evolutionary mechanisms in CASs; when applied to the interpretation of CNs, they can be characterized as internal processes of recovery and challenges to a regional network following an external shock. This aspect of network resilience is more process-oriented, in contrast to the structure and function that directly reflect the state at a given moment. How to interpret the true resilience of a network system requires a process of system adaptation and recovery. Thus, these three mechanisms can be used to analyze the resilience of a system in terms of its ability to withstand risk and recovering more effectively.

- Tagging is the process of aggregation formation, tagging facilitates the identification and selection of different subjects or targets and selective interactions. The network size, i.e., the number of nodes and connections, is used to describe the corresponding network characteristics, which can provide a good explanation of the selection behavior of subjects when networks are aggregated.
- Building blocks are the main feature of the internal mechanism, i.e., they are closely related to diversity, which generates complexity due to the diverse combinations which exist within a CAS. The combination of blocks will be changed by an agent in response to new circumstances. Using connectivity to represent the building blocks of a network system reveals that networks are composed of interconnected combinations of internal nodes and nodes; and the greater the connectivity, the greater the diversity, i.e., the more complex the internal mechanism.

- Internal models represent the internal structure of a subject through which the environment and behavior of that subject can be inferred. The subject relies on its complex, unique internal structure and can accumulate experience and learn or predict certain things to demonstrate that each subject in the system has its own complex internal mechanisms. The ability to diffuse factor flows in urban networks is portrayed using transmissibility, which is related to the shortest path between nodes. Higher transmissibility means that the urban nodes in a network can achieve faster exchanges of factors such as information, knowledge, and capital, which promotes inter-city learning and innovation and enhances the resistance of a region to crises. In response to shocks, paths with fewer hops are more reliable, and at the same time, can respond to external changes more rapidly and cope with disruptions more smoothly. Quantitative assessments of network transportability using the metric of network efficiency are directly based on the transport functions achieved by the network. They can also better portray the internal mechanisms of complex networks.

## 3. Materials and Methods

A technical diagram was derived based on the concept and mechanism of urban network resilience, consisting of three main steps. The first step is to build the population mobility network. We constructed the network in different periods using Baidu migration data, which describes the characteristics of the general network. In the next step, seven network typological indicators were selected to measure the NR. The third step involved exploring the attribution of NR. A spatial econometric model was used to analyze the factors contributing to NR and reveal its evolution processes (Figure 2). Furthermore, dominant and vulnerable nodes could be identified to enhance the regional network structure and improve the sustainability of the integrated area.

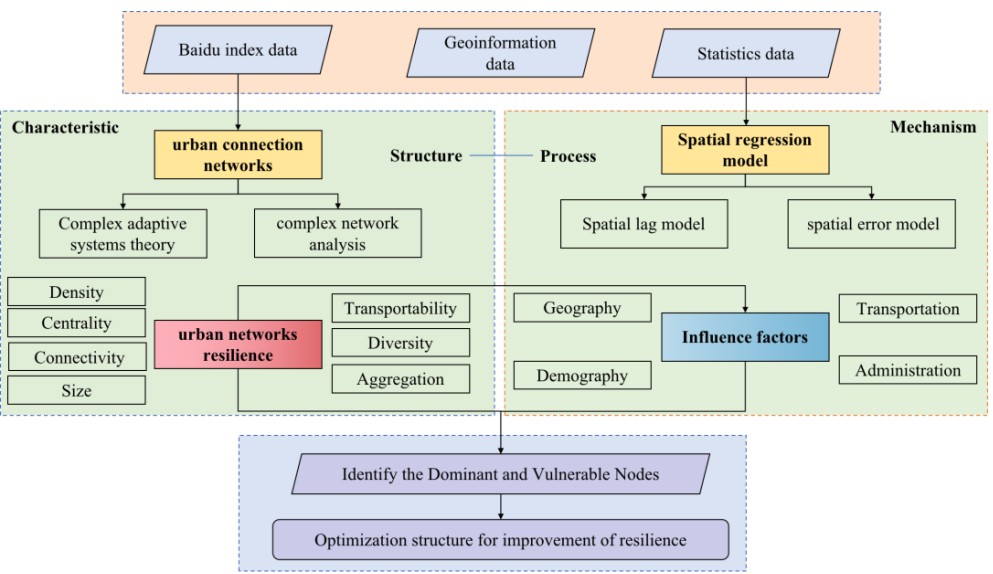

**Figure 2.** Integrated framework of network resilience for population mobility.

### 3.1. Study Area

The Yangtze River Delta region (YRDR) is located in the east of China (Figure 3). It is one of the largest urban agglomerations in the world, comprised of Jiangsu, Zhejiang, and Anhui provinces, as well as the Shanghai municipal government. With a population of 0.227 billion people in 2019 and an area of 358,000 square kilometers, this region accounts for 3.69% of China's total land area. Approximately 1/6 of China's population lives in this area, and nearly 1/4 of the national economic output is derived from it. It is one of the most densely populated and economically developed areas in the country. The urbanization rate in this region reached nearly 75% in 2020.

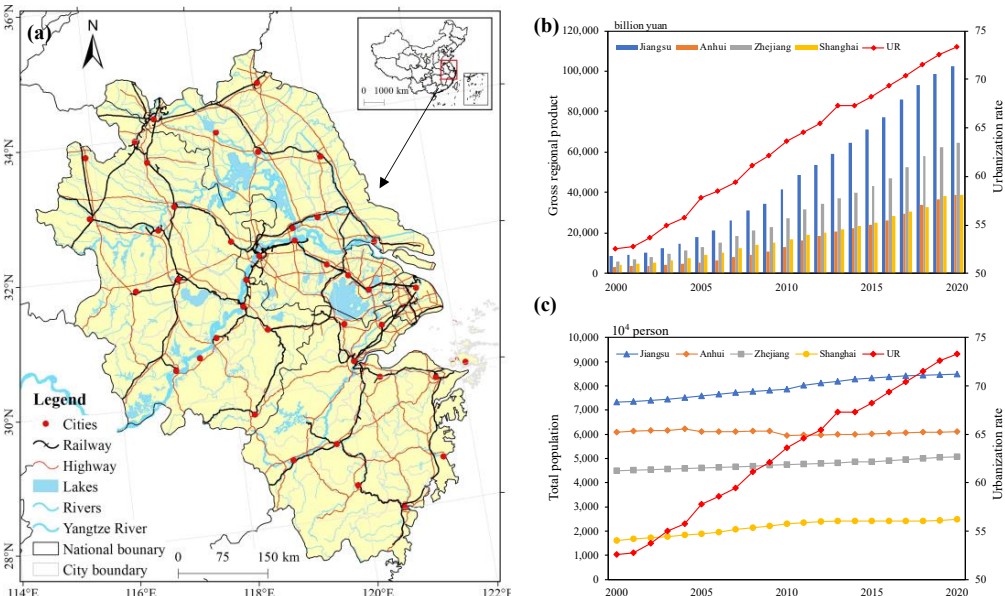

**Figure 3.** The Yangtze River Delta region. (**a**) The study area; (**b**) Population and urban ratio change from 2000 to 2020; (**c**) GDP and urban ratio change from 2000 to 2020.

With the most active economic development, the highest degree of opening to the world, and the most substantial innovation ability, YRDR plays an essential strategic role in national modernization in China. Since 1983, it has been exploring how to break down administrative boundaries and promote intercity cooperation. Yangtze River Delta integrated development has been promoted as a national strategy, promulgating the policy of "Outline of the Regional Integration Development Plan of the Yangtze River Delta" since 2019. This had made the region a focal point for socio-economic development and has considerably increased the rate of urbanization. As the most integrated region in China, the YRDR has apparent variations among the different scales. There has been a rapid flow of population, technology, capital, transportation, and tourism, resulting in the formation of a polycentric and flat network space [32,33]. However, due to the impact of COVID-19, inter-city mobility has declined significantly, and some connections have been "weakened" or even "interrupted". The urban network structure has also been significantly impacted by these disruptions.

### 3.2. Data Sources

(1) Population mobility data. These data come from the Baidu Migration Dataset, provided by the Baidu Huiyan platform (https://huiyan.baidu.com/products/platform, accessed on 20 January 2022), which records the intensity of population mobility between any two cities and visualizes movement during a specific period [34]. Although it cannot capture all the migratory population based on the availability of smartphones, when using big data to analyze the spatial characteristics of population mobility, dimensionless measures and relative indicators of the data are better than those reflected by absolute values [35]. We collected the daily population mobility data of 41 cites in the YRDR from 19 January to 27 March 2021 and defined a differential ratio indicator to compare the incoming and outgoing population using the following equation:

$$r = \left( \sum_{city=i}^{N} \left| \frac{in_{numcity}}{out_{numcity}} - 1 \right| \right) / N \tag{1}$$

where $in_{numcity}$ and $out_{numcity}$ are a city's incoming and outgoing populations and $N$ is the number of cities. Generally, the ratio of $in_{numcity}$ to $out_{numcity}$ should be near 1, and $r$

should be near 0. Whenever incoming and outgoing flows of cities are not balanced, *r* is significantly higher than 0.

According to the "The General Office of the State Council on the arrangement of holidays in 2021", the Spring Festival holiday took place from 11–17 February, a total of 7 days, in 2021. As the travel period, we selected days when the daily passenger flow was higher than the average and the difference was greater than the average. Spring Festival transport data can be divided into three periods (Figure 4). The first concerns the return period (3–10 February), when most people return to their hometowns for family reunions. From 11–17 February, the Spring Festival period, the inter-city population fluctuated to some extent. The second is the leaving period (from 18–28 February), when people leave their hometowns to return to the city where they currently live. The "ordinary" period from 12–24 March was selected for comparison. Due to the impact of the resurgence of COVID 19 across the country before 2021, the total mobility before the Spring Festival was significantly less than what was observed subsequently; this was mainly related to the local policy proposed at that time. Therefore, it was appropriate that we chose this period to analyze the characteristic of population mobility under a disruption simulation. The data we used were the sum of daily population movements in each city during the three phases mentioned above, which illustrated trends in inter-city movements during this time frame.

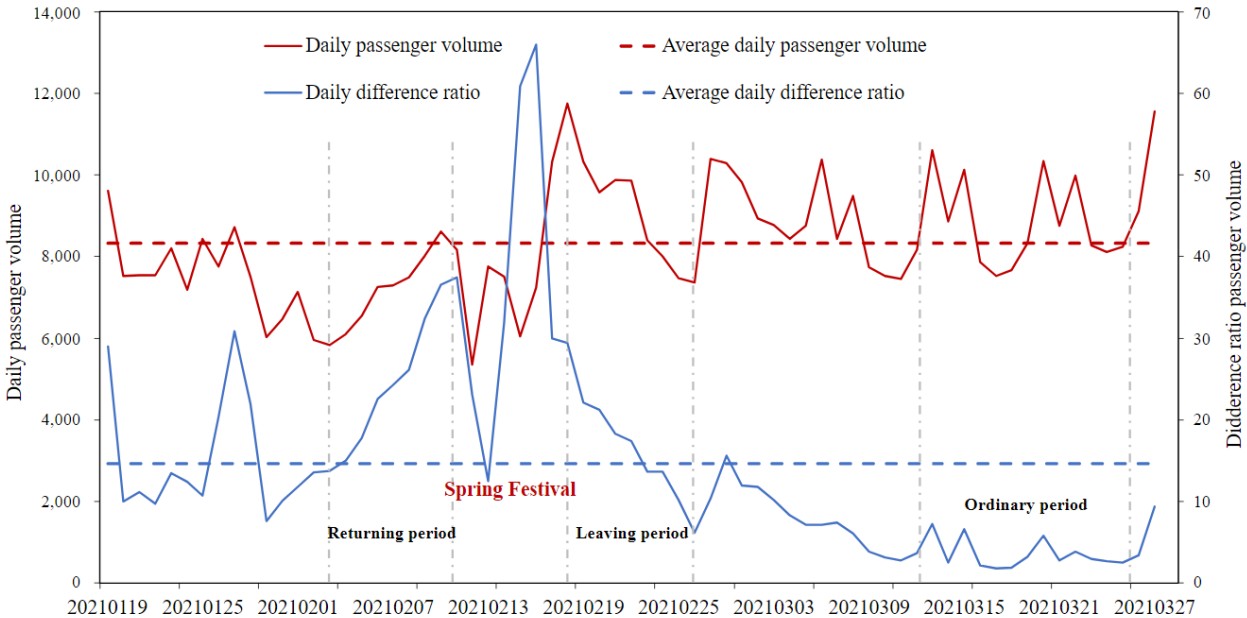

**Figure 4.** Passenger volume from January to March during the Spring Festival in 2021.

(2) Social statistical data. Socio-economic statistics and the transportation and tourism data of each city, including GDP, per capita GDP, every year resident population, tertiary industry share data, employment data, and road mileage data of each city, were obtained from the provincial and municipal statistical yearbooks and statistical bulletins in 2021.

(3) Geographic Information Data. The administrative district boundaries and city administrative centers were obtained from the National Basic Geographic Information Center database at a resolution of 1:4 million (https://www.resdc.cn/, accessed on 22 February 2022). All maps were made from the standard map with review No.GS (2020) 4619, downloaded from the standard service website of the National Bureau of Surveying and Mapping Geographic Information. The distances between cities were calculated using the latitudes and longitudes of urban administrative centers.

### 3.3. Methods

#### 3.3.1. Assessment of Network Resilience

The structures of population migration patterns, with significant correlation and complexity, can be abstracted as a network. Each city is regarded as a node, and the intensity of population migration can be summarized as the directed line edge weights between nodes. The overall structural characteristics of the network are generally investigated using nodes, density, average path lengths, and clustering coefficients. The individual structure features of the network are typically characterized by centrality, transmissibility, and diversity. To reveal and illustrate the features of the network's topological structure and the statistical properties of population migration, the following network analysis indicators were selected:

$$R(x) = f(D, C, L, S, E, V, A) \tag{2}$$

where $R(x)$ is a function of network resilience; $D$ represents the characteristic of consistency, characterized by network density; $C$ represents the characteristic of centrality, which is represented by the degree of centrality; $L$ represents the connectivity feature of the network, measured by the average of independent paths; $S$ denotes the size of the network, as represented by the total number of nodes and edges; network efficiency ($E$) is used to measure network transmissibility; the average number of independent paths ($V$) is used to measure the network diversity; and finally, the average clustering coefficient ($A$) represents the network aggregation. The specific calculation formula is listed in Table 1.

**Table 1.** Measuring the index of network resilience.

| Characteristic | Indicators | Formula | Descriptions |
|---|---|---|---|
| Density | Network density (D) | $D = \sum_{i=1}^{n} \sum_{j=1}^{n} \frac{d(n_i, n_j)}{n(n-1)}$ | $n$ is the number of nodes in the network, $d$ is the intensity of population mobility |
| Centrality | Degree centrality (C) | $C = K_i / N - 1$ | $K_i$ is the degree of node $i$, and $N$ is the number of nodes in the network |
| Connectivity | Average network distance (L) | $L = \frac{1}{1/2n(n+1)} \sum_{i \neq j} d_{ij}$ | $d_{ij}$ is the degree of node $i$, and $N$ is the number of nodes in the network |
| Scale | Network size (S) | $S = n + M$ | $M$ represents the total number of edges, $n$ is the number of nodes in the network |
| Transmissibility | Network efficiency (E) | $E = \frac{1}{n(n-1)} \sum_{i \neq j} \frac{1}{D_{ij}}$ | $D_{ij}$ is the shortest path length from node $i$ to $j$ |
| Diversity | Average number of independent paths (V) | $V = \sum_{i \neq j} n_{ij} / n(n-1)$ | $n_{ij}$ is the number of independent paths between node $i$ and node $j$ |
| Aggregation | Average clustering coefficient (A) | $A = 1/N \sum_{i}^{n} \frac{2E_i}{k_i(k_i-1)}$ | $E_i$ is the number of paths between a node and its $K_i$ neighboring nodes |

NR is calculated quantitatively using the resilience formula proposed by Dixit et al. [20]. Additionally, we modified the formula to calculate the NR of population mobility based on the relationship between seven topology indicators, as shown in (3):

$$R = L \times S \times E \times V / D \times C \times A \tag{3}$$

Using the method of interrupt simulation, changes in network resilience may be described. The role of each city in the overall network was obtained by comparing our simulation to real results.

For the quantitative calculation of anti-interference of network resilience, we simulated changes in the network resilience under disruption due to node failure. Node failure considers impacts, such as an epidemic, on different cities. As targets of the attack, cities are removed, one at a time. This means that a change in the overall network reflects the dynamic processes which determine its degree of resilience, as shown in (4):

$$R_p = R_a - R_b \tag{4}$$

where $R_p$ represents the change of network resilience and $R_a$ and $R_b$ are the network resilience before and after node interruption.

### 3.3.2. Scenarios of Nodes Disruption

The robustness and resilience of a network are critical to risk reduction and loss mitigation. Network resilience describes the ability of a system to maintain its essential functions in the face of internal disruptions or external changes. To further explore the network resilience of population mobility, we simulated the disruption of connections in cities under the impact of a disruption and observed the changes in their topologies. Two simulation methods expressed the interaction between overall network resilience and city node failures.

It is possible for the failure of a node within a network to result in the removal of that node from the original network. This can result in a fundamental change in the characteristics of the network. Natural disasters such as typhoons, tsunamis, and earthquakes, as well as random events such as urban isolation caused by public health emergencies such as epidemics, can be used to simulate the impact of external shocks on nodes and network connections. Thus, we set the scenario as "**node failure**". In this scenario, nodes are ranked according to their degree of centrality, and failures are simulated sequentially. These urban nodes are removed from the network in turn to form a new network structure, and the topological metrics of the network are calculated.

However, in the "node failure" scenario, only the impact of a specific node city's failure on the overall network may be known, so it is not possible to quantify the importance of the node city or the potential to recover to the original level of network resilience. Changes in network resilience following the failure of different node cities can be attributed to that node city's ability to withstand external shocks. According to our theoretical framework, after a system is hit by an external shock, its internal resistance and defense are enhanced through adaptive mechanisms. A stable equilibrium is achieved through recovery and reorganization. Network resilience and recovery must be distinguished from the resilience of specific features and functions. As a result, we set another scenario, i.e., "**network recovery**", and calculated the network indexes of different nodes after failure as the resilience state, $S_1$, of the corresponding nodes. By comparing $S_1$ with $S_0$, we could determine the impact of urban failure on $S_1$.

Comparing these two scenarios, the essential difference was that in a "node failure" scenario, node cities disappeared after an attack, and therefore, the overall network evolved and changed. In contrast, although the node cities in "network recovery" scenario were different for each node failure and the network was dynamic, the overall number of connections remained unchanged. By comparing the relative amount of change in resilience before and after the failure, the inherent resistance and resilience of the network became the focus.

The essential difference between these two scenarios is that in Scenario 1, node cities disappear after an attack, resulting in a change in the overall network, while the node cities in Scenario 2 vary each time they fail and the network is dynamic; as such, the overall number of connections remains the same. The disruption scenario simulation was designed to explore the resistance and resilience of the network by comparing the relative changes in resilience before and after a disruption.

### 3.3.3. Spatial Econometric Regression Model

(1) Variable selection. We selected network resilience (NR) in the three periods of the Spring Festival in 2021 as the dependent variable. According to the gravity model, the strength of inter-city connections is related to the attractive scale of the destination city (economic scale, population size, etc.) and the cost of transportation (such as transportation connectivity, travel time, etc.). After reviewing existing research, 10-factor explanations for the resilience of the urban network were selected. GDP can be interpreted as the measure of economic development of urban areas. The rate of urbanization and the proportion of tertiary industry can serve as measures of urbanization. The total population and total number of employees represent the scale of urban population and employment. The abundance of tourism resources represents the prevalence of urban tourism. Road,

Transport and Car represent the transportation infrastructure conditions. After logarithmic processing, all variables could be used to eliminate dimensional differences (Table 2).

**Table 2.** Variable selection and descriptive statistics.

| Type | Name | Abbreviations | Mean | Std. Dev. | Min | Max |
|---|---|---|---|---|---|---|
| Independent variable | Changes in network resilience | NR | 8.142 | 0.078 | 8.071 | 8.485 |
| | | | 8.146 | 0.077 | 8.054 | 8.452 |
| | | | 8.131 | 0.083 | 8.067 | 8.550 |
| Dependent variable | Intensity of population mobility [1] | Mobility | 7.894 | 0.661 | 6.595 | 9.386 |
| | Intensity of population mobility [2] | | 6.944 | 0.752 | 5.638 | 8.709 |
| | Intensity of population mobility [3] | | 6.997 | 0.547 | 5.471 | 8.012 |
| | Gross domestic product | GDP | 8.266 | 0.918 | 6.746 | 10.564 |
| | Total population at the end of the year | POP | 6.134 | 0.677 | 4.753 | 7.819 |
| | Urbanization rate | UR | 0.664 | 0.113 | 0.420 | 0.893 |
| | Total number of employees | Lab | 4.877 | 1.232 | 1.515 | 6.617 |
| | Proportion of tertiary production | Is | 0.523 | 0.085 | 0.423 | 0.883 |
| | Highway mileage | Road | 9.357 | 0.484 | 7.565 | 10.130 |
| | Passenger transport volume | Transport | 7.862 | 0.874 | 6.161 | 9.951 |
| | Motor vehicle ownership | Car | 4.591 | 0.856 | 3.012 | 6.112 |
| | Number of domestic tourists | Tourist | 8.331 | 0.810 | 6.923 | 10.069 |

Intensity of population mobility [1–3] during the return, leaving, and ordinary periods, respectively.

(2) Spatial model. Considering the spatial dependence of intercity mobility, we used a spatial econometric model to explain the influencing factors and detect the spatial effects of network resilience in the YRDR during the Spring Festival. The spatial error model (SEM) and the spatial lag model (SLM) are principally used to describe spatial correlations [36]. The SLM supposes that the spatially averaged weight of the adjacent *NR* partially reduces the value of NR observed in city *i* due to spatial interactions. The SLM is expressed as:

$$lnNR_{i,t} = \rho w_{i,t} lnNR_{i,t} + \beta_i x_{i,t} + \mu_i + \gamma_t + \varepsilon_{i,t} \tag{5}$$

The SEM integrates spatial relationships based on the spatial dependence between the error terms associated with local and neighboring cities. The SEM is defined as:

$$lnNR_{i,t} = \beta_i x_{i,t} + \mu_i + \gamma_t + \varepsilon_{i,t}, \ \varepsilon_{i,t} = \lambda w_{ij} \varepsilon_t + \mu_{i,t} \tag{6}$$

where $lnNR_{i,t}$ stands for the *NR* of city *i* at time *t*; $x_{i,t}$ is the explanatory variable; $\beta_i$ is the coefficient to be estimated; $\rho_i$ is the coefficient of spatial autoregressive; $\lambda$ is the coefficient of spatial error; $\varepsilon_t$ indicates the impact of the shock on neighboring cities; $\mu_i$ and $\gamma_t$ are the individual and time effects; and $\mu_{i,t}$ is random error.

## 4. Results

### 4.1. Spatial Patterns within Population Mobility Networks

We constructed connection networks and linkages based on the intensity of population mobility among cities. The degree of centrality of each node was further analyzed, as this reflects the radiation effect of a city node on others in the network. In our study, a city is considered a regional central city with a high degree of centrality, strong convergence or evacuation ability, and high communication ability. The spatial pattern of each connection network and node degree centrality is presented in Figure 5.

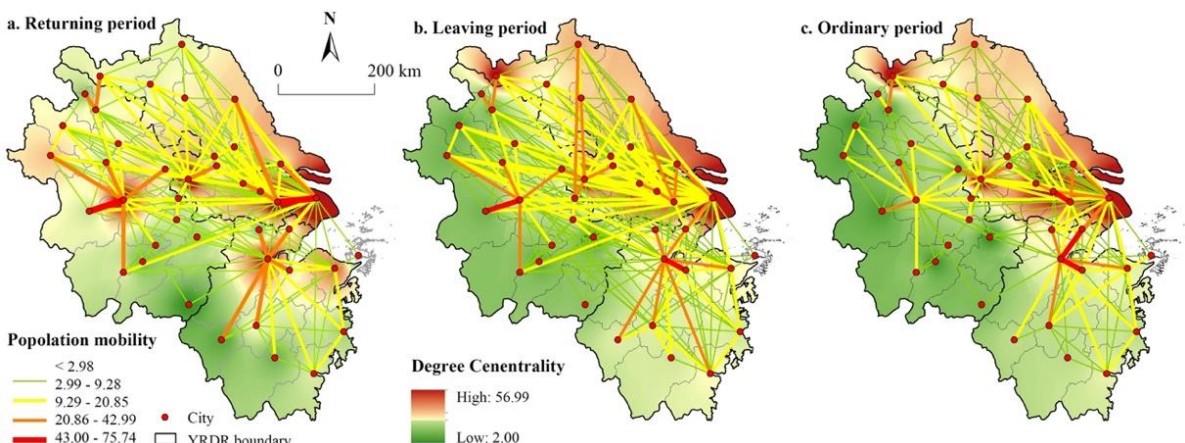

**Figure 5.** Spatial patterns of population flow networks and node degree centrality in the Yangtze River Delta Region.

The intensity of population flow between cities at the first level (43.00–75.74) presented scattering. Strong linkages existed between the core city and its adjacent nodes, such as Shanghai–Suzhou, and Hefei–Lv'an. The second level (20.86–42.99) presented an axis of Shanghai–Suzhou–Wuxi–Changzhou and two clusters around Hangzhou and Hefei. The third level (9.29–20.85) reflected the association between provincial capital cities and other cities within the same province. About 83% of all city linkages were at level four (<2.98), indicating that the intensity of most cities is low, and most city connections are relatively fixed. In the ordinary period, strong connections between cities in terms of daily population flow are mainly regional in nature.

During the return period, the top 10 cities were as follows: Shanghai, Nanjing, Suzhou, Hefei, Hangzhou, Wuxi, Ningbo, Changzhou, Yangzhou, and Fuyang. Spatially, these cities formed several obvious regional high-value agglomeration areas. In the leaving period, cities in Jiangsu Province were high-value areas for population mobility. The top 10 cities were as follows: Shanghai, Suzhou, Nanjing, Hangzhou, Hefei, Wuxi, Ningbo, Fuyang, Changzhou, and Yangzhou. The provincial capital cities became the nodes with the greatest population flow during this period. At the same time, some traffic node cities also had a higher degree of centrality than before. In the ordinary period, which was roughly the same as the leaving period spatial pattern type, the difference was that the degree of interconnections between cities had weakened.

### 4.2. Assessment and Characteristics of Network Resilience

Changes in the network topology indexes were diverse in the different periods, so the network resilience tended to be stable over time. As shown in Table 3, the centrality and connectivity of the overall network gradually decreased, while the density, size, and aggregation of the network first increased and then decreased, presenting an inverted "V" trend. The average daily passenger volume in the return and leaving periods was much higher than usual.

The centrality and connectivity of the network were enhanced when mobility intensity was high and tended to be flat when liquidity decreased. This could be verified with the spatial variation characteristics of the previous network connection pattern. The network density, size, and aggregation should change, because the increase in population mobility during the Spring Festival leads to more paths with similar distances. This linkage did not reveal strong flow intensity, and the spatial pattern did not significantly increase in terms of the number of strong connections. However, it changed the overall network topology and increased network redundancy, impacting network resilience.

**Table 3.** Network structure resilience.

| Topological Index | ND | NC | NL | NS | NA | NE | NV | NR |
|---|---|---|---|---|---|---|---|---|
| Return period | 0.214 | 0.684 | 2.084 | 392 | 0.476 | 0.763 | 4.929 | 4672.070 |
| Leaving period | 0.231 | 0.661 | 2.004 | 420 | 0.518 | 0.740 | 4.332 | 3997.759 |
| Ordinary period | 0.210 | 0.614 | 1.964 | 389 | 0.476 | 0.824 | 4.530 | 3970.058 |
| Trend | ⌄ | ⌄ | ⌄ | ⌃ | ⌃ | ⌄ | ⌄ | ⌄ |

*ND* is the network density; *NC* is the network degree centrality; *NL* is the network connectivity; *NS* is the network size; *NA* is the average clustering coefficient; *E* is the network efficiency; *NV* is the average number of independent passageways; *NR* is network resilience.

In addition, the transmissibility and diversity of the network decreased first and then increased, but notably, network transmissibility was the highest in the ordinary period, i.e., it increased from 0.763 to 0.824. The diversity index decreased from 4.929 to 4.530 at the end of the Spring Festival. The changes in network diversity were mostly caused by the decrease of population flow through a decline of physical connections, thus obviously leading to the deterioration of the fault tolerance of the network. While there was a lot of population movement during the Spring Festival, this also encouraged connections to the city, which translated into a slightly higher than usual network connectivity. When a path was interrupted, other paths could ensure the normal function, thus effectively maintaining the stable operation of the network. Overall, the level of network toughness gradually stabilized from 4672.07 in the return period to 3970.058, with no significant change in the leaving and ordinary periods.

*4.3. Network Resilience Changes under Disruption Scenarios*

4.3.1. Changes in Overall Network Resilience: "Node Failure"

Figure 6a–g shows the proportion of failed city nodes and the changes in the network topology index. The network topology indexes showed different trends in three time periods as the node failure rate increased. Specifically, network density (ND) and network size (NS) are linked to the nodes and edges, so node failure will cause them to decrease. Overall, both indices decrease as the node failure rate increases. As the proportion of intentional attacks increases, network centrality (NC) progressively decreases. During the return period, when the node failure rate was 0.8, NC was reduced to 0 and the network failed. In the ordinary period, the whole network failed after 0.9, implying that the anti-attack ability of network centrality is more robust in the "normal" period than during the Spring Festival.

The network connectivity (NL) and diversity (NV) indices initially increased and then decreased as the degree of node failure increased. Both achieved the highest values following the failure of some nodes in connection with the network. The change in network efficiency (NE) also confirmed this, with the difference being that NE increased first after node failure decreased. When the attack rate reached about 70%, the network suddenly failed and dropped to 0. Network aggregation (NA) decreased and increased, becoming invalid in the end.

Figure 6h illustrates how the NR changes in a node failure scenario. As network nodes fail, the NR increases accordingly. With 20% of nodes failing during the leaving period, the NR reached the highest level, while in the other two periods, the NR was 5% and 12%. Clearly, the NR maintained a relatively stable rate during the return period. However, the network collapse was accelerated when nodes failed, as it had greater volatility during the leaving and ordinary periods.

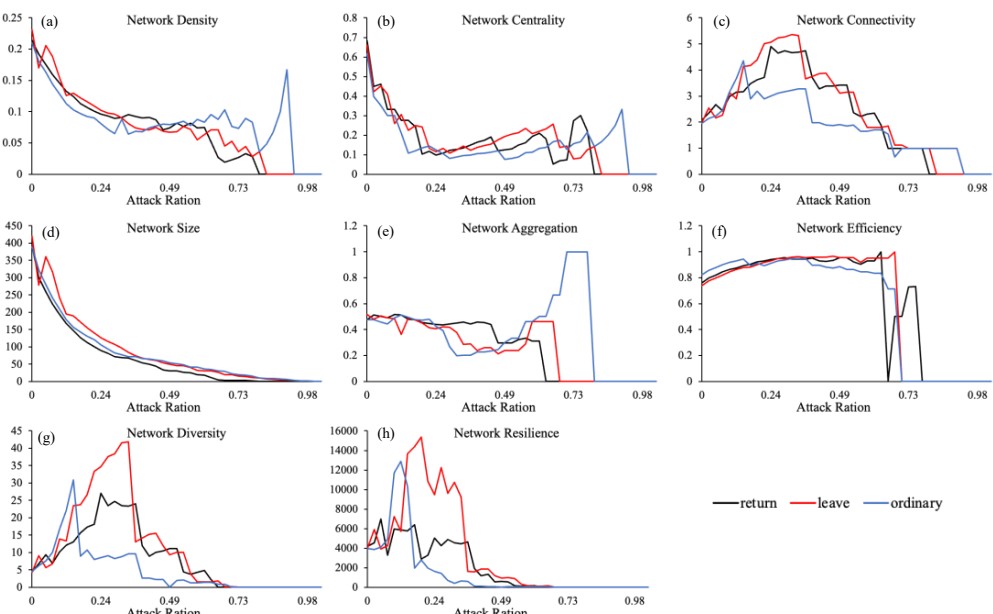

**Figure 6.** Changes in complex network topological indexes under different disruption scenarios.

### 4.3.2. Changes in the Resilience of Node Cities: "Network Recovery"

The second simulation method considered the impact of COVID-19 on different city nodes. We simulated the threat of a public emergency event using the nodes within the network as the attack object, as well as the interruption of the network every time a node was lost. The design involved simulated attacks on 41 nodes. Likewise, the topology indicators and NR were updated. Afterwards, each node was removed or isolated until all the nodes in the network had been removed.

A positive correlation exists between urban nodes and centrality in the YRDR, showing the spatial characteristics of the agglomeration along the transport corridor. Network resilience (NR) and degree of city centrality (NC) were divided into five levels, as shown in Figure 7. First- and second-level cities, such as Shanghai, Suzhou, Wuxi, Nanjing, and Hefei, were primarily provincial capitals and regional core cities, forming a distribution trend along two distinct axes: Shanghai–Nanjing and Shanghai–Hangzhou. With the increase in population flow, this phenomenon became more pronounced, most notably in the return period. Thus, development along the Yangtze River Economic Belt axes is key to achieving the sustainable network in the YRDR.

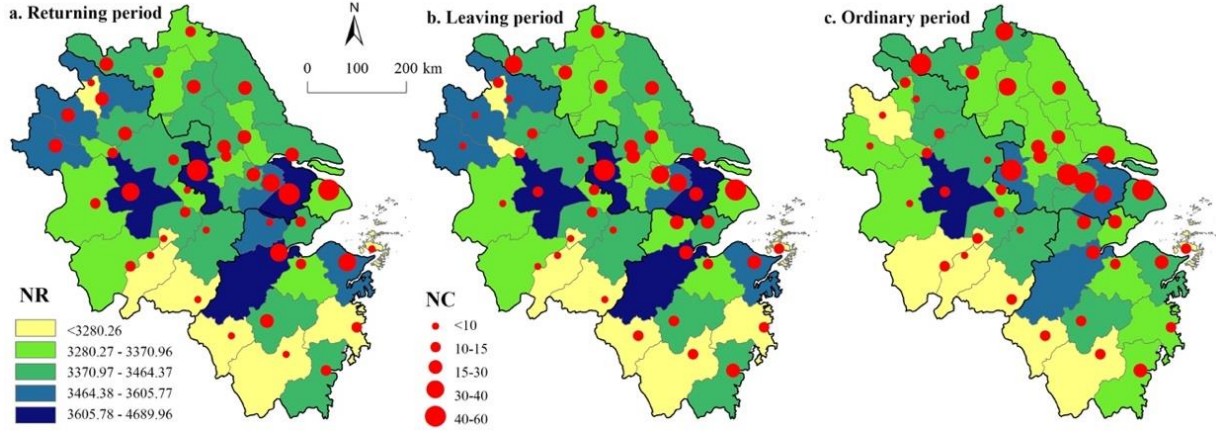

**Figure 7.** Spatial pattern of the impact of urban node failure on network resilience.

### 4.3.3. Variation of Node Anti-Interference of Network Resilience

Furthermore, different city nodes present different levels of anti-interference between NRs, which means that the failure of other nodes weakens the resilience of city nodes. However, only one city typically had the highest levels of resilience when the resilience of each city was examined under a state of maximum disturbance. As shown in Figure 8, with the interruption of network nodes, the attenuation degree of node resilience presented differences. During the return period, Huaibei, Zhoushan, Chizhou, Quzhou, and Lishu experienced the greatest declines, while Nanjing, Hefei, and Hangzhou saw relatively little change. The spatial distribution pattern decreased from the periphery to the center. The situation during the leaving period was similar. Due to the increase in population mobility, peripheral cities increased their network resilience and were highly vulnerable to the impact of node failures. In the ordinary period, this phenomenon was more prominent: the attenuation of network resilience decreased from outside to inside, and the core cities demonstrated the least volatility.

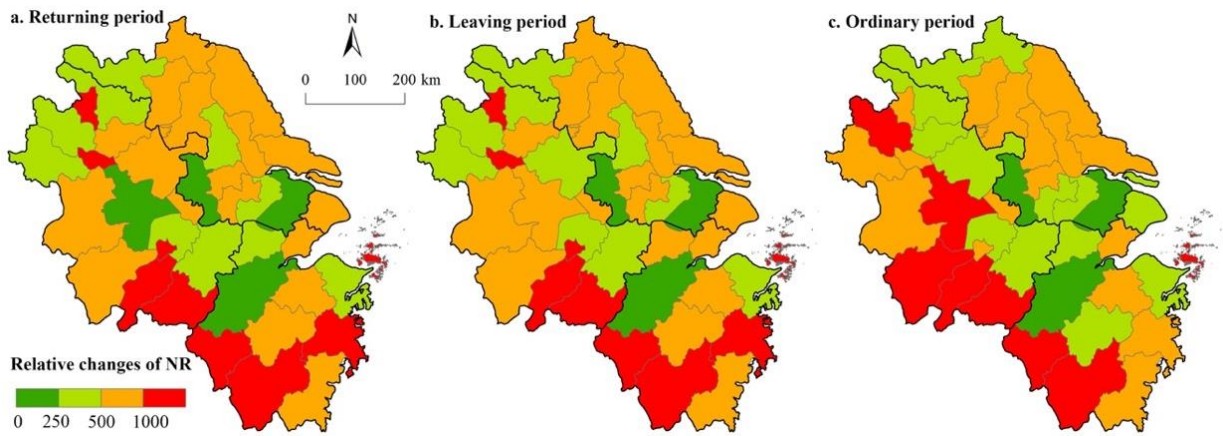

**Figure 8.** Spatial pattern of attenuation in node resilience.

Another crucial node related to resilience in the YRDR is called the "vulnerability node". The resilience of cities is likely to be significantly attenuated by the failure of other cities. Vulnerable nodes reduce the resilience of the overall network. The spatial distribution of vulnerability nodes also showed agglomeration characteristics. Most nodes with significant declines were in the Anhui and Zhejiang provinces, forming agglomeration areas with apparent vulnerabilities in the periphery. The cities in Jiangsu Province and Shanghai demonstrated higher fault tolerance and a more robust ability to resist shock.

### 4.4. The Mechanism of Urban Networks Resilience

We selected and analyzed factors influencing the resilience of migration networks in different periods. Firstly, a collinearity test was conducted to eliminate variables with multicollinearity. Finally, eight independent variables, i.e., GDP, number of employees, urbanization rate, administrative level, road density, aviation, high-speed rail, and tourism resource richness, were retained. According to the collinearity diagnosis results, the final independent variable variance inflation factor (VIF) was less than five, and tolerance was greater than 0.30, indicating no obvious collinearity between the respective variables. By comparing the R2, logarithmic likelihood, AIC, and SC values of the three models (Table 4), it was found that the SLM model had the most significant logarithmic likelihood and the smallest AIC and SC values, making it suitable for identifying factors which influence intercity travel.

**Table 4.** Spatial econometric model results for network resilience.

| Variables | OLS | | SLM | | SEM | |
|---|---|---|---|---|---|---|
| | Coefficient | Z Value | Coefficient | Z Value | Coefficient | Z Value |
| W_NR | | | −0.029 *** | −3.783 | | |
| Lambda | | | | | −0.399 * | −1.618 |
| Mobility | 0.060 ** | 2.195 | 0.084 *** | 4.036 | 0.068 *** | 3.226 |
| GDP | −0.138 | −2.589 | −0.178 *** | −4.440 | −0.146 *** | −3.367 |
| POP | −0.020 | −0.340 | −0.022 | −0.497 | −0.018 | −0.374 |
| UR | 0.570 ** | 2.551 | 0.692 *** | 4.201 | 0.608 *** | 3.435 |
| Lab | 0.038 ** | 2.731 | 0.044 *** | 4.241 | 0.035 *** | 3.266 |
| Is | 0.105 | 0.528 | 0.128 | 0.888 | 0.134 | 0.813 |
| Road | 0.056 *** | 1.835 | 0.104 *** | 4.092 | 0.104 ** | 2.370 |
| Transport | 0.026 | 1.417 | 0.006 | 0.427 | 0.023 | 1.517 |
| Car | 0.046 | 1.472 | 0.085 *** | 3.427 | 0.043 | 1.627 |
| Tourist | 0.034 * | 1.625 | 0.032 ** | 2.115 | 0.029 ** | 1.840 |
| Constant | 7.088 *** | 22.167 | 6.905 *** | 29.216 | 7.083 *** | 25.535 |
| $R^2$ | 0.685 | | 0.766 | | 0.701 | |
| LogL | 70.620 | | 76.774 | | 71.087 | |
| AIC | −117.241 | | −127.547 | | −118.174 | |
| SC | −96.678 | | −105.271 | | −97.611 | |

W_NR is the coefficient of spatial autoregressive; Lambda is the coefficient of spatial error. The symbols ***, **, and * denote the significance at 1%, 5%, and 10%, respectively.

The spatial lag coefficient of W_NR was −0.029; this result passed the 99% significance test, indicating that urban network resilience has noticeable spatial spillover effects. The degrees of network resilience of adjacent cities influence each other. Overall, if the resilience of a node city increases by 1%, those of the surrounding adjoining node cities will decrease by 0.029%. This shows that network resilience has strong regional linkages, and the cascading effect between nodes reduces the overall resilience.

According to the results of the SLM, five independent variables, i.e., the mobility of the urban population (mobility), urbanization rate (UR), the total number of employees (lab), road density (road), and the tourism (tourist), passed the significant level test, and all were positively correlated with urban resilience. The scale of urban population flow (mobility) indicated the population migration situation during the period; the higher the value, the greater the interconnection between cities. When mobility between cities increased, the resilience of the city network was the highest. The coefficient of the urbanization rate (UR) was the largest at 0.570; this result passed the 5% significance test. Cities with higher urbanization levels have advantages in terms of infrastructure and emergency support and have played a significant role in enhancing urban resilience. The total number of employees (lab) reflects the labor force in each city. The return of the labor force after the Spring Festival is the main reason for population migration. Its coefficient was 0.038; this result has passed the 5% significance level test. The increase in the number of laborers returning to their hometowns during the Spring Festival also improved the ability of the network to cope with more uncertain risks. The "road" and "tourist" variables passed the 1% and 10% significance level tests, respectively.

On the one hand, this shows the essential supporting role of intercity transportation connectivity. Traveling abroad during the Spring Festival holiday will also affect network resilience. On the other hand, the impact of this is relatively small, indicating that people will reconsider travel risks due to restrictions against the background of the normalization of epidemic prevention and control.

## 5. Discussion

### 5.1. Identifying Dominant and Vulnerable Nodes

It is important to understand how the failure of a dominant and vulnerable city node in the YRDR affects the resilience of the network structure. The process of identifying

this node is based on the comprehensive distribution of NR over different time periods. Four dominant nodes and seven vulnerable nodes, each heavily influencing the network structural resilience in the YRDR, were found. Nanjing, Suzhou, Hangzhou, and Hefei are the four dominant node cities with high levels of centrality and control. Each of these has a relatively high degree of economic development and a comprehensive transport center.

There were seven vulnerable nodes (Huaibei, Tongling, Huangshan, Quzhou, Lishui, Taizhou, and Zhoushan) that showed specific agglomeration distribution characteristics in space, i.e., mostly peripheral or border cities in the YRDR. Spatial aggregation leads to a regional lock-in effect. When a node reduces in scope, it does not significantly impact the larger regional environment. This could result in insufficient resource replenishment, unbalanced supply and demand, and uneven development in small areas. Most such nodes were far away from provincial capital cities or at the junctions between provinces. Limited by inadequate administrative barriers and transport infrastructure, such cities may become "dead cities".

### 5.2. Optimization for Improvement of Resilience

When a dominant node is paralyzed during a crisis, it interferes tremendously with the resilience of the network structure. Meanwhile, vulnerable nodes are the best indicators of network resilience. Uncertainty about the global spread of COVID-19 persists. Various phenomena have made us realize that the coordinated management of public health emergencies in the YRDR still needs to be strengthened [37]. Therefore, we propose a cross-scale collaborative spatial governance system, the "Region-Metropolitan-City" (Figure 9). Through multi-level linkages, this approach can address the uncertain disturbances caused by the network cascading effect and improve the resilience of a regional network.

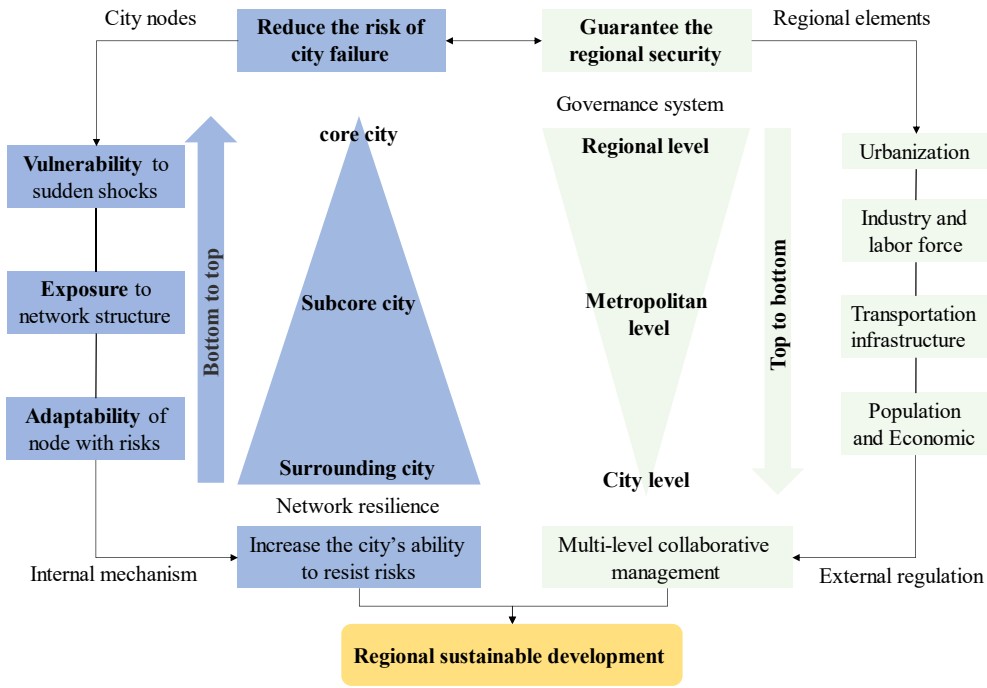

**Figure 9.** Optimization of network resilience framework.

Diverse and frequent elemental flows can lead to regional perturbations or collapses due to the strengthening of regional negative externalities. Current territorial spatial planning focuses on urban resilience; its attention to regional resilience is by no means sufficient. Regional networks need to demonstrate resilience to external disruptions to ensure regional coordination, stability, and sustainable development [38,39]. Hence, the regional integration of various elements (urbanization, industry, transportation, infrastructure, and

socioeconomic) and different scenarios (disruption and external shocks) should be considered in future spatial planning. The failure probability of node cities can be reduced by increasing the abilities of nodes to resist risks. At the same time, by building a cross-scale collaborative management system to ensure the overall security, this integrated approach will greatly enhance the resilience of city nodes and regions. Therefore, node improvement and system optimization are two important pathways to enhance regional resilience.

*5.3. Limitations and Further Works*

While we have analyzed network resilience from the perspectives of CAS and CN, there are still some limitations that could be further investigated, including the following: We were not able to obtain population mobility data in the YRDR for 2020 or 2022, so this paper could not verify the difference in the impact of the COVID-19 on urban network resilience through comparisons with historical periods. We only simulated the spatial characteristics of regional network resilience under disruption based on data, which may have some effects on our policy implications. Due to the suddenness and randomness of public health emergencies, we did not consider random combinations of different city failures. Rather, we just simulated changes in network resilience under the cumulative failure of each node city, along with the decline in resilience under each node failure, to better understand which cities were the most vulnerable points and how they could be managed more effectively. Additionally, we did not consider how network resilience changes under different combinations of city failures, which should be the subject of a future study. Our objectives were to introduce a complex perspective for the evaluation of network resilience and to assess whether neighboring nodes have a mutual influence on network resilience. As a result, the groundwork has been laid for the next combination of scenarios and network cascade effects under random disruption in different cities within the network.

The promulgation of the Yangtze River Delta Integration Policy has injected new vitality into regional interconnectivity and social-economic development. We could consider incorporating regional resilience improvement and vulnerability governance in future planning practices, which could be improved in the following ways: (1) through exhaustive studies of the risk response capabilities of non-core cities and quantitative analyses of other modes of population flow when non-core cities are disturbed; (2) by comprehensively assessing changes in the resilience of population mobility networks and the spread of epidemics, disaster risks, and government control measures; and (3) by discussing in more detail how long it takes the system to recover from a disturbed state to a steady state and the degree to which different policies influence this transition.

## 6. Conclusions

This paper proposes a conceptual framework for network resilience using complex adaptive systems in combination with complex network analysis. In contrast with previous studies, the theoretical model was applied to an actual analysis through a network resilience evaluation of population mobility in the Yangtze River Delta region, demonstrating the rationality of the theoretical assumptions. Using node failure simulation and spatial effect analysis, we also proposed a governance strategy to optimize and improve network resilience in the region. The major conclusions are as follows:

First, the intensity of travel during the Spring Festival in 2021 had obvious characteristics over time. Due to restrictions related to the COVID-19 pandemic, the population mobility before the holiday was less than after the holiday. Second, the network resilience in the YRDR was greatly affected by its topological characteristics, which are closely related to urban connections. The network structure was found to be unstable, and the interruption caused responsiveness and resilience to synchronous decline. NR showed a dependence on transportation corridors, and the urban nodes that significantly interfere with the overall resilience of the network structure were mainly concentrated on axes that were consistent with the Shanghai–Nanjing and Shanghai–Hangzhou development axes. Third, disruption

simulations can be used to further identify critical elements that affect the resilience of network structures. With the failure of node cities, network resilience first rises and then declines; in the YRDR, population mobility has a certain degree of redundancy. Finally, the network resilience in the YRDR has a negative spatial spillover effect. The factors which affected NR during the Spring Festival included urban scale such as urbanization rate and labor force, as well as traffic connectivity. Tourism attractiveness and population size gradually decreased as a result of the COVID-19 pandemic. Generally, the most effective means to increase the resilience of regional networks are the acceleration of urbanization and the enhancement of local transportation infrastructure.

Overall, our research contributions are mainly reflected in the following aspects: at the methodological level, we propose a method by which to assess regional network resilience using disruption scenario simulations and spatial effects analyses to identify vulnerabilities and key influencing factors; at the theoretical level, we present a theoretical framework for measuring regional complex network resilience, which enhances the potential of theoretical analyses of regional and urban network resilience. Specifically, our method treats network resilience as a system structure, rather than as a collection of characteristics and relationships. The findings of this study also demonstrate that via the construction of a cross-scale collaborative spatial governance system, uncertainty disturbances caused by network cascading effects can be resolved and insights can be gained regarding the sustainability of other regions.

**Author Contributions:** Conceptualization, L.C.; methodology, L.C. and Y.Z.; software, L.C. and Y.Z.; validation, H.C. and D.S.; formal analysis, Y.Z.; investigation, J.X., X.Y. and Z.P.; resources, L.C. and Y.Z.; data curation, B.W.; writing—original draft preparation, L.C.; writing—review and editing, H.C.; visualization, L.C. and Y.Z.; supervision, H.C.; project administration, B.W.; funding acquisition, H.C. and B.W. All authors have read and agreed to the published version of the manuscript.

**Funding:** This research was supported by the Class A Strategic Pioneer Science and Technology Special Project of the Chinese Academy of Sciences (XDA19040502), the Foundation of Key Talent Projects of Gansu Province (No. 2021RCXM073) and the National Natural Science Foundation of China (Grant Nos. 52178043).

**Institutional Review Board Statement:** Not applicable.

**Informed Consent Statement:** Not applicable.

**Data Availability Statement:** The data presented in this study are available on request from the corresponding author.

**Conflicts of Interest:** The authors declare that they have no known competing financial interest or personal relationship that could have appeared to influence the work reported in this paper.

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
