# Peer review of "Evaluation of the Spatial Effect of Network Resilience in the Yangtze River Delta: An Integrated Framework for Regional Collaboration and Governance under Disruption"

_land, doi:10.3390/land11081359_

Round 1
Reviewer 1 Report
The paper addresses a relevant and current issue, the spatial effects of resilience networks in China´s region. The author's proposal is to use CAS to evaluate network resilience. Nevertheless, a more deep demonstration must be pointed out. Figure 1 must be more explicit, namely with the integration of several issues related to structure resilience and the issues of process resilience in order to better understand the reader.
Another point is related to the use of mobility (daily? Commuting?) and migration (for a how long period? No daily, we supposed) data. They have completely different patterns, so a differential use will be required.
When scenarios are discussed, a deeper explanation of each one must be clarified in order to better evaluate the network answer, namely to understand figure 6.
These are the general considerations. Some others are more specific. In line 165, what is the percentage of the population in order to compare with are and economic output?
In lines 183/184 What is the meaning of a high-speed network of the population?
Lines 247 to 251 and Table 1, Connectivity and Diversity are different but have the same indicator designation.
Concerning results, they are very interesting, but the paper must be improved. Consider splitting this paper into two different ones, in order to be more focused. One more methodological and other more result analytic.
Author Response
Dear Reviewer 1,
Thank you very much for your valuable comments and suggestions on our manuscript. Following the your comments, we have modified and improved our manuscript according to your kind advice and detailed suggestions. Enclosed please find the responses to the referees. We sincerely hope this manuscript will be acceptable to be published in the Special Issue “Regional Sustainable Development of Yangtze River Delta, China” on the Land.
Point-by-point responses to all the reviewers’ comments are in the attached document.
Thank you very much for all your help and looking forward to hearing from you soon.
Best regards

Reviewer 2 Report
- It is not quite clear the relation of theoretical hypothesis and the expected practical improvement the paper suggest
- In l.119 I recommend to reformulate “interact in complex ways” in “interact in autonomous diverse ways” (the complexity is supposed to be caused by this uncoordinated diversity. Or not?)
- However, complexity should be defined in terms different from diversity or anyothers used in theoretical part. Otherwise there is no real theory/ hypothesis but a tautology. An insightful definition is needed.
- Network resilience should be clarified. What does it happen in case of lack of resilience?
- In l.197: “41 cites” means 41 networks? What are these “cites”? Of how many nodes (cities)? Were the 41 cites randomly selected?
- In l. 541-2, “Due to the 541 suddenness and randomness of public health emergencies, we have not considered the 542 random combination of different city failures,” needs clarification
- The conclusions are more descriptive (like a summary). They should become more insightful pointing out the innovativeness of the results at theoretical and/or practical level.
- In table s1, the phrase “…are relatively independent” is confusing. Isn’t something either dependent or independent (at from an everyday point of view)? Clarify
- In table s1 “aggregation” both as a characteristic and index is confusing. This is also the case for “diversity”
Author Response
Dear Reviewer 2,
Thank you very much for your valuable comments and suggestions on our manuscript. Following the your comments, we have modified and improved our manuscript according to your kind advice and the detailed suggestions. Enclosed please find the responses to the referees. We sincerely hope this manuscript will be acceptable to be published in the Special Issue “Regional Sustainable Development of Yangtze River Delta, China” on the Land.
Point-by-point responses to all the reviewers’ comments are in the attached document.
Thank you very much for all your help and looking forward to hearing from you soon.
Best regards

Round 2
Reviewer 1 Report
The quality of the article has improved significantly. Congratulations. Just a few minor proposed changes.
In line 112 CAS and CN are referred to for the first time, so you should give their meaning.
In line 314 migratory population is mentioned I suggest the use of mobile population.
In line 356 could you mention the referential geographical system (is not mandatory, but it helps)?
Line 412, in fact, is recalculated.
Lines 429 and 430 could be clearer about the meaning of the period's designation, returning, leaving, and ordinary.
Author Response
Dear reviewer,
We thank you for your constructive evaluation and helpful comments on our article. We have modified the text taking account into the your suggestions. Point-by-point responses to you comments in the attachment.
Sincerely Yours
